# Current Evidence Supporting the Role of Immune Response in ATTRv Amyloidosis

**DOI:** 10.3390/cells12192383

**Published:** 2023-09-29

**Authors:** Domenico Plantone, Guido Primiano, Delia Righi, Angela Romano, Marco Luigetti, Nicola De Stefano

**Affiliations:** 1Department of Medicine, Surgery and Neuroscience, University of Siena, 53100 Siena, Italy; d.righi4@student.unisi.it (D.R.); destefano@unisi.it (N.D.S.); 2Dipartimento di Neuroscienze, Organi di Senso e Torace, Fondazione Policlinico Universitario Agostino Gemelli IRCCS, 00168 Rome, Italy; guidoalessandro.primiano@policlinicogemelli.it (G.P.); angela.romano12@gmail.com (A.R.); mluigetti@gmail.com (M.L.); 3Dipartimento di Neuroscienze, Università Cattolica del Sacro Cuore, 00168 Rome, Italy

**Keywords:** hereditary transthyretin amyloidosis, polyneuropathy, familial amyloid polyneuropathy, inflammation

## Abstract

Hereditary transthyretin (ATTRv) amyloidosis with polyneuropathy, also known as familial amyloid polyneuropathy (FAP), represents a progressive, heterogeneous, severe, and multisystemic disease caused by pathogenic variants in the *TTR* gene. This autosomal-dominant neurogenetic disorder has an adult onset with variable penetrance and an inconstant phenotype, even among subjects carrying the same mutation. Historically, ATTRv amyloidosis has been viewed as a non-inflammatory disease, mainly due to the absence of any mononuclear cell infiltration in ex vivo tissues; nevertheless, a role of inflammation in its pathogenesis has been recently highlighted. The immune response may be involved in the development and progression of the disease. Fibrillary TTR species bind to the receptor for advanced glycation end products (RAGE), probably activating the nuclear factor κB (NF-κB) pathway. Moreover, peripheral blood levels of several cytokines, including interferon (IFN)-gamma, IFN-alpha, IL-6, IL-7, and IL-33, are altered in the course of the disease. This review summarizes the current evidence supporting the role of the immune response in ATTRv amyloidosis, from the pathological mechanisms to the possible therapeutic implications.

## 1. Introduction

Hereditary transthyretin (ATTRv) amyloidosis with polyneuropathy, also known as familial amyloid polyneuropathy (FAP), represents a progressive, heterogeneous, severe, and multisystemic disease caused by pathogenic variants in the *TTR* gene, which is ultimately fatal [1,2]. This autosomal-dominant neurogenetic disorder has an adult onset with variable penetrance and an inconstant phenotype, even among subjects carrying the same variants [3,4]. Being a rare disease, ATTRv amyloidosis with polyneuropathy has a global prevalence of 5500–38,500, and is considered endemic in a few countries, including Portugal, Sweden, and Japan [5].

Lacking any gross evidence of inflammation in ex vivo tissue specimens, ATTRv amyloidosis has been classically considered a non-inflammatory disease. Nevertheless, there is growing evidence in the literature regarding some role of inflammatory mechanisms underlying this rare disorder [6,7,8]. It has been postulated that the immune response may participate in the disease’s development and progression; recent studies are exploring this hypothesis, with interesting results.

This review will summarize the current evidence supporting the role of the immune response in ATTRv amyloidosis, from the pathological mechanisms to the possible therapeutic implications.

## 2. Materials and Methods

To select the relevant literature for this narrative review, the authors first searched the PubMed, Embase, and Google Scholar databases using the following string: “familial amyloid polyneuropathy” OR “hereditary transthyretin amyloidosis with polyneuropathy” AND (“inflammation” OR “immune response” OR “cytokine” OR “lymphocytes”). Articles, in languages other than English, published before the 1980s, or reported only as abstracts, were not considered suitable and were discarded unless considered relevant. The authors then evaluated the abstracts of all of the articles and selected those focused on the aim of the present narrative review. The reference lists of all of the selected articles were also evaluated to identify additional relevant articles. After this phase, 49 articles were selected in total.

The literature search was performed by DP, and then all of the authors independently evaluated the relevance of each article on a scale from 0 (not relevant) to 5 (very relevant). A consensus was then reached by all authors to include only those publications with a combined score of at least 3 points. For clinical papers, the relevance of each study was based on the number of subjects studied, the accuracy of the diagnostic work-up used, the methods used to assess the immune response, and the clinical scales included in the study. Meanwhile, for pre-clinical papers, the relevance was based on the robustness of the experimental methods and their translatability to the clinical setting. After this phase, 17 papers were selected. Among these, 7 were prospective studies on humans, 1 was a clinical trial, 2 were case reports, and 8 were studies on animal models (one study included investigations on humans and animal models, and was therefore counted twice). Three studies reached a score of 3, and the rest of them reached a score of 4 (Figure 1).

## 3. Hereditary Transthyretin Amyloidosis with Polyneuropathy: Current Pathogenetic View

Transthyretin (TTR), previously known as prealbumin, is a serum and cerebrospinal fluid protein that is synthesized by the liver, by the retinal epithelium, by the pancreas [9], and by the choroid plexus in the brain; it represents a transport protein for thyroxine and retinol-binding proteins associated with vitamin A [10,11,12]. It forms a tetrameric protein made up of four identical subunits, and has a total molecular mass of 55 kDa [13]. The stability of the tetrameric structure is undermined by pathogenic variants in the TTR gene, which produce misfolded monomers that accumulate in extracellular spaces and progressively aggregate in oligomers, and ultimately into amyloid fibrils with the cross β-sheet structure [3].

ATTRv amyloidosis is generally regarded as a multisystem disorder, with the peripheral nervous system and heart being the two main sites of involvement [14]. The gastrointestinal tract, kidneys, and eyes can also be significantly involved in amyloid deposition [1,15]. While the toxic processes may differ in the peripheral nerve and the other sites of involvement, the systemic inflammatory response may likely overlap. The deposition of TTR amyloid fibrils seems to be related to the severity of the disease [16]. Polyneuropathy represents the most common of the ATTRv amyloidosis phenotypes, and is most frequently associated with Val30Met mutation. It is characterized by an early onset, a high penetrance, and a progressive axonal length-dependent injury [17,18]. Nonetheless, electrophysiological [19,20] and pathological [4,21,22] studies documented possible myelin abnormalities, together with axonal loss and, interestingly, these myelin alterations are usually in close contact with TTR deposits, suggesting their direct effect. 

An extensive review of all the possible clinical features of ATTRv amyloidosis goes beyond the scope of the present article, and can be found elsewhere [23,24]. Patients with ATTRv amyloidosis may experience several neurological and non-neurological symptoms, which may reduce their quality of life [25,26].

The sensory and autonomic symptoms are due to the peripheral nervous system’s involvement with pain [27], paresthesia and hypoesthesia, bilateral carpal tunnel syndrome, digestive disorders, erectile dysfunction, postural hypotension, fatigue, and weight loss [1,15,28]. Cardiac abnormalities, including hypertrophy, arrhythmias, ventricular blocks, and cardiomyopathy [29], together with renal abnormalities, including albuminuria and mild azotemia, and vitreous opacities, complete the symptomatological picture [30]. ATTRv amyloidosis polyneuropathy has a rapidly progressive course compared to other conditions that may clinically present similarly [1]. Due to the rarity of the disease, the diagnosis can be challenging, with a high rate of misdiagnosis, particularly in non-endemic areas, and with consequent delays in treatment initiation [28]. The therapeutic scenario of ATTR amyloidosis has been completely revolutionized in recent years [31,32]. Liver transplantation was the first therapeutic option available [33], and a shorter disease duration is the main factor associated with its better outcome [34,35]. Diflunisal [36] and tafamidis [37] are TTR protein stabilizers that can delay the progression of the disease if started promptly in its early phases. In addition, patisiran [38], inotersen [39,40], and vutrisiran [41] are three *TTR* gene-silencing medications that significantly decrease TTR production, leading to clinical stabilization or slight improvement.

How amyloid deposits induce neurodegeneration in ATTRv amyloidosis is still a matter of debate, and different hypotheses have been proposed. The first theory postulates a direct mechanical effect of the aggregates on the nerve fibers, given the spatial coincidence of axonal and myelin injury, and amyloid polymers [42,43,44]. The second hypothesis proposes that amyloid deposits trigger significant oxidative stress with toxic lipid peroxidation and subsequent neuronal and myelin injury [45]. Interestingly, the high susceptibility of unmyelinated postganglionic autonomic nerve fibers to oxidative stress may explain the predominant neuronal loss in sympathetic ganglia in early-onset ATTRv amyloidosis patients, in which amyloid fibrils are usually longer and thicker compared to late-onset patients [46,47]. However, regarding the different toxicity of TTR polymers and TTR oligomers, the majority of the tissue culture studies show that TTR oligomers, not the fibrils, are the toxic elements in any target organ, and in vivo studies in humans report the appearance of non-fibrillar deposits in both peripheral nerves of early V30M patients, as well as in human wtTTR transgenic mice showing non-fibrillar cardiac deposits before fibril deposition [48]. Nerve ischemia, caused by perivascular amyloids, has also been proposed as a possible mechanism for nerve damage in ATTRv polyneuropathy [49]. A further possible explanation refers to the impairment of the cross-talk between Schwann cells and axons, with reduced neuronal trophism and ultimately neuronal loss [50]. Moreover, the mechanism of apoptosis has been implicated in the pathogenesis of FAP. Fas is a receptor capable of initiating apoptosis through the activation of caspase 8, and both of these proteins have been described as activated in affected patient tissues. Interestingly, the activation of caspase 8 and the absence of changes related to Bcl-2, Apaf-1, Bax, and caspase 9 suggest that FAP is characterized by a mitochondrion-independent apoptosis [51].

Finally, the last but perhaps the most fascinating theory that explains the axonal and myelin damage in ATTRv polyneuropathy involves the inflammatory response induced by ATTRv amyloid deposits that may significantly contribute to progressive nerve injury. However, it should also be made clear that the evidence for inflammation in humans with ATTRv amyloidosis is not substantial, and its effect is also seen as more related to amyloid toxicity with secondary inflammatory changes, rather than a primary event. Clear clinical signs of inflammation or the immune response, such as elevated C-reactive protein or the presence of inflammatory infiltrate in biopsies, are lacking in humans. Therefore, the actual role of inflammation in ATTRv polyneuropathy is still under debate, and convincing evidence for inflammation or the immune response in human ATTRv amyloidosis patients is still lacking.

Nevertheless, taking into account all of the possible pathogenetic hypotheses previously described, it must be admitted that none of them necessarily eliminates the immune inflammatory response as playing a role in the development of the disease. In fact, any or all of them could induce or provide the stimulus for immune activation.

In the following paragraphs, we will present the current evidence supporting the role of the immune response in ATTRv.

## 4. Evidence of Immune Response in ATTRv Amyloidosis

The contribution of the immune response to the ATTRv amyloidosis pathogenesis has been highlighted only in the last two decades, although its precise contribution is still far from being characterized in detail, and represents a matter of research. Figure 2 summarizes the possible role of the immune response in ATTRv amyloidosis.

### 4.1. Human Studies: State of the Field

Table 1 summarizes the main studies exploring the contributions of the immune response in patients with ATTRv polyneuropathy.

The first consistent evidence came from a study performed more than 20 years ago by Sousa and colleagues, which demonstrated that fibrillary TTR species bind to the receptor for advanced glycation end products (RAGE) [52]. The authors evaluated the expression of RAGE and TTR deposition in the affected tissues, and found that FAP patients had higher levels of RAGE expression compared to controls, with the distribution of immunoreactive TTR appearing to overlap with that of RAGE. This was confirmed also in peripheral nerve sections of FAP patients. Moreover, they demonstrated that the binding of TTR fibrils to RAGE results in the activation and nuclear translocation of NF-kB, which is very important for regulating immune and inflammatory responses [52]. Matsunaga and colleagues studied gastrointestinal tract autopsy samples in FAP patients, and found that the distribution of RAGE and AGE strongly correlate to that of amyloid deposits, with a clear relationship between TTR, AGE, and RAGE, but no correlation between NF-kB, apoptotic marker, and amyloid deposits [53]. When interpreting these results, we should highlight that the Matsunaga study evaluated histological samples prepared from the gastrointestinal system (stomach, small and large intestines), whereas peripheral neurons and lung tissue of FAP patients were evaluated in Sousa studies. The gastrointestinal tract lacks myelinated nerves, and unmyelinated fibers are thought to be more resistant to compression from amyloid deposits than myelinated fibers [54]. This may at least partly account for the discrepancy between these studies.

There is an ongoing debate on the non-specificity of RAGE binding across many neurologic disorders and different forms of amyloid fibrils, and it is still not clear from the literature if this is TTR polyneuropathy-specific. In fact, published data suggest that the interaction of RAGE with fibrils may be coincident with the toxic response, independent of tissue damage. RAGE represents a member of the multiligand cell surface immunoglobulin family [55] and is expressed by different cells, including endothelial and smooth muscle cells, dendritic cells, lymphocytes, neutrophils, monocytes, and macrophages [56]. The pivotal role of RAGE has already been documented in the pathogenesis of multiple neurological diseases including Alzheimer’s disease, amyotrophic lateral sclerosis, Parkinson’s disease, and Huntington’s disease [57]. Remarkably, RAGE has other ligands than AGE, including amphoterin [58], S100/calgranulins [59], and amyloids forming β-sheet fibrils [60,61]. Relevantly, the role of RAGE has also been investigated in the pathogenesis and progression of peripheral neuropathy. By binding its ligand s100B, which is secreted by the Schwann cells, RAGE mediates peripheral nerve repair in vivo [62]. However, RAGE, together with its pro-inflammatory ligands, has been found to be overexpressed in human diabetic nerves, and the activation of the RAGE–NF-κB-dependent pathway seems to significantly contribute to the progression of the disease [63]. In further support of this evidence, it should also be remembered that both the AGE and RAGE levels measured in skin biopsies strongly correlate with neuropathy severity [64,65]. Finally, Yan and colleagues in a mouse model of systemic amyloid A (AA) amyloidosis, demonstrated that RAGE also binds amyloid A, and that by antagonizing RAGE, NF-κB activation is suppressed, as well as the expression of proinflammatory cytokines and the accumulation of amyloid fibrils [66]. From this perspective, a recent study found decreased S100A8 plasma levels in ATTRV30M patients, as well as a dysregulated S100 expression by Schwann cells in response to TTRV30M, and by mutated macrophages in response to Toll-like receptor agonists [67]. Taking into account the data from the literature, we would be inclined to define the role of RAGE as non-ATTRv-specific.

NF-κB activation induces two major signaling pathways, the first called the “canonical” and the second called the “noncanonical” pathway [68]. Several molecules, including various cytokines, are involved in the activation of pattern-recognition receptors (PRRs); TNF receptor (TNFR) superfamily members and T and B-cell receptors activate the first pathway, which results in IκBα degradation in the proteasome, induced by its site-specific phosphorylation by a multi-subunit IκB kinase (IKK) complex [69]. This results in the transient nuclear translocation of the p50/RelA and p50/c-Rel dimers [70], which are crucial for the activation of the immune response, including for the production of IFN-alpha, IFN-beta, [68,71], and for Th1 cell differentiation [71].

The noncanonical pathway is activated by a different group of ligands, including lymphotoxin β receptor, B-cell activating factor receptor, CD40, and receptor activator nuclear factor-kappaB (RANK), and involves the NF-κB2 precursor protein, p100 [72]. The noncanonical NF-κB pathway can be viewed as a supplementary signaling axis that cooperates with the canonical NF-κB pathway in the regulation of specific functions of the adaptive immune system [73].

Related to the activation of NF-kB, some studies have documented an increased expression of TNF-α, macrophage colony-stimulating factor, and IL-1β in the endoneurial axons of FAP patients, since the earlier stages of the disease and a progressive increase in their levels are associated with the ongoing neurodegenerative process [52].

Few studies explored the serum inflammatory profiles of FAP patients. Azavedo and colleagues evaluated the serum levels of TNF-α, IL-1β, IL-6, IL-8, IL-33, IFN-β, IL-10, IL-12, and cortisol in 28 Brazilian ATTRv patients and found increased levels of TNF-α, IL-1β, IL-8, IL-33, IFN-β, and IL-10, and decreased levels of IL-12 in comparison to healthy age-matched subjects [7]. Interestingly, these authors found that serum levels of IL-33, IL-1β, and IL-10 were already high in asymptomatic patients, hypothesizing that inflammation started even before fibril deposition [7]. Another similar study was performed by our group on 16 ATTRv patients and 25 healthy controls. We found that serum levels of IFN-alpha and IFN-gamma were higher, whereas IL-7 levels were lower, in ATTRv patients. However, no significant difference between groups was found regarding IL-1Ra, IL-6, IL-2, IL-4, and IL-33 levels, and no correlation was found between IFN-α, IFN-γ, IL-7, and clinical severity [8]. Remarkably, the lack of confirmation in relation to the higher levels of IL-33 levels in our ATTRv patients was attributed to the fact that the majority of patients recruited by Azavedo and colleagues were asymptomatic and mildly symptomatic, with IL-33, a nuclear alarmin, released from the nuclei of producing cells after the injury [74]. In light of these data, we can hypothesize that the release of this cytokine becomes less abundant with the progression of the disease [8].

In contrast with these two studies, a previous study [6] documented an increased serum concentration of IL-6 in FAP carriers and patients. The reason for this discrepancy in the results was explained by the strict age-dependence of IL-6 levels and the well-known circadian rhythm of IL-6 secretion in both young and older persons, [75] with both conditions potentially influencing the results.

The source of the production of these inflammatory molecules is still a matter of debate without any definitive evidence [7,8]. It is still unknown whether the inflammatory changes in FAP patients primarily relate to the pathogenesis of the disease, or are a secondary response to tissue injury. Furthermore, no data are available on the circulating peripheral blood mononuclear cell modifications in ATTRv patients [8].

Kurian and colleagues studied the global peripheral blood cell mRNA expression profiles from 263 tafamidis-treated and untreated V30M FAP patients, from asymptomatic V30M carriers, and from a group of healthy controls [76]. They found that 1426 genes associated with 36 significant pathways were differently expressed in symptomatic subjects. The 18% of them mapped to immune and inflammatory processes. Remarkably, the eukaryotic initiation factor-2 (eIF2) pathway was downregulated in all symptomatic subjects, as well as primary immunodeficiency signaling, and purine nucleotide biosynthesis. On the contrary, they found that signaling networks for FCγ, triggering receptor expressed on myeloid cells 1 (TREM1), natural killer (NK) cells, and cytokines IL3, IL15, and IL22 were all upregulated in the FAP patients. They also found a significant difference in the different expressions of several genes between the symptomatic and asymptomatic female subjects, that somehow could be related to an earlier onset and more progressive disease course in the male V30M FAP patients. In detail, symptomatic females showed a downregulation of eIF2, primary immunodeficiency, T-helper cell differentiation, and inducible T-cell costimulator (iCOS) signaling pathways. On the contrary, in symptomatic males, 29 significant canonical pathways linked to immunity, including Fcγ receptor, NK cell, TLR, B-cell receptor, leukocyte extravasation, and IL-12 signaling, were all upregulated. Finally, the authors demonstrated a trend toward the normalization of all of these altered gene expressions in patients treated with tafamidis [76].

Further data of enormous interest finally arrived from a recent publication from Moreira and colleagues [67]. They found that plasma levels of the S100A8 protein were lower in ATTR V30M patients compared to healthy controls. Furthermore, they found that S100A8/9 levels in Schwann cells were dysregulated after incubation with human V30M TTR and by mutated bone marrow-derived macrophages in response to Toll-like receptors agonists.

Furthermore, given the still not fully defined significance of the immunological changes documented in ATTRv amyloidosis, it should be acknowledged that there are data supporting a potentially protective role of the immune response against disease progression. In this regard, a recent report described three male patients with ATTR amyloidosis cardiomyopathy-associated heart failure that resolved spontaneously, with reversion to near-normal cardiac structure and function [77]. In each of the patients, high-titer circulating polyclonal IgG antibodies against human ATTR amyloids were identified, which were documented to bind specifically to ATTR amyloids. The authors demonstrated that these antibodies were not present in a group of 350 other consecutive patients with ATTR amyloidosis cardiomyopathy experiencing a typical clinical course. Therefore, a possible role of these antibodies in the reversal of the disease could be hypothesized [77]. Moreover, other authors reported TTR-specific B-cells from humans without ATTR amyloidosis [78,79]. These observations reinforce the hypothesis that these antibodies could be involved in the removal of amyloids in the presymptomatic phase of the disease in some subjects not having ATTR at all or having subclinical amyloids. This evidence suggests that this B-cell response is likely secondary rather than primary in the disease pathogenesis. Interestingly, Michalon and colleagues identified NI301A, a monoclonal antibody that selectively binds with high affinity to the disease-associated ATTR aggregates of either the wild-type or ATTRv-related disease. This process occurred through the isolation of a collection of ATTR-binding antibodies selected for high-affinity binding to ATTR, amyloid-removal activity, and absent binding to naturally folded TTR, by screening human memory B-cell libraries derived from healthy elderly subjects [78].

Analyzing the studies available so far in the literature, two main limitations are clear: the absence of longitudinal studies that outline the course of the inflammatory response over the course of the disease, and the small number of patients studied, due to the low frequency of FAP in the population and the lack of multicenter studies.

**Table 1 cells-12-02383-t001:** Summary of selected studies on ATTRv patients evaluating the immune response modifications.

Reference	Methods and Techniques	Main Findings
Sousa et al., 2001 [52]	Analysis of nerve biopsy samples from patients by semiquantitative immunohistology and in situ hybridization	Increased levels of RAGE beginning at the earliest stages of the disease; upregulation of TNF-α, IL1- β, and iNOS in a distribution overlapping RAGE expression.
Matsunaga et al., 2002 [53]	IHC and sequential IF staining	RAGE and AGE have a distribution strongly correlated to that of amyloid deposits. However, no correlation was detected between NF-κB, apoptotic marker, and amyloid deposits.
Azevedo et al., 2019 [7]	ELISA	Increased serum levels of TNF-α, IL-1β, IL-8, IL-33, IFN-β and IL-10, and decreased levels of IL-12 in ATTRv patients.
Luigetti M et al., 2022 [8]	Luminex XMAP multiplexing technology	Increased serum levels of IFN-alpha and IFN-gamma, and decreased serum levels of IL-7 in ATTRv patients.
Suenaga et al., 2017 [6]	ELISA, cell culture, and Bio-Plex pro cytokine assay kit	IL-6 serum concentration was elevated in FAP carriers. In native TTR culture conditions, IL-6 increased in CD14 + monocytes in the presence of V30M-mutated TTR, compared with wild-type TTR, in a TTR-dose-dependent manner. IL-6 concentration increased in CD4 + T cells and CD8 + T cells in a TTR-dose-dependent manner. IL-1β, TNF-α, and IL-10 increased in a TTR-dose-dependent manner in CD14 + monocytes.
Kurian et al., 2016 [76]	Microarray technology and Luminex bead assays	Downregulation of eIF2 pathway in all symptomatic subjects, as well as primary immunodeficiency signaling, and purine nucleotide biosynthesis. Signaling networks for FCγ, TREM1, NK cells, IL3, IL15, and IL22 were all upregulated in FAP patients. Symptomatic females showed a downregulation of eIF2, primary immunodeficiency, T-helper cell differentiation, and iCOS signaling pathways. In symptomatic males, 29 significant canonical pathways linked to immunity, including Fcγ receptor, NK cell, Toll-like receptor, B-cell receptor, leukocyte etravasation, and IL-12 signaling, were all upregulated. There was a trend towards the normalization of all these altered gene expressions in patients treated with tafamidis.
Moreira et al., 2023 [67]	Real-time PCR, cell culture	Plasma levels of S100A8 protein were lower in ATTR V30M patients compared to healthy controls; S100A8/9 levels in Schwann cells were dysregulated after incubation with human V30M TTR and by mutated bone marrow-derived macrophages in response to Toll-like receptor agonists.

AGE, advanced glycation end products; ATTRv, transthyretin amyloidosis; CD, cluster of differentiation; eIF2, eukaryotic initiation factor-2; ELISA, enzyme-linked immunosorbent assays; FAP, familial amyloid polyneuropathy; iCOS, inducible T-cell costimulator; IF, immunofluorescence; IFN, interferon; IHC, immunohistochemistry; IL, interleukin; iNOS, inducible form of nitric oxide synthase; MCSF, macrophage colony-stimulating factor; NF-kB nuclear factor kappaB; NK, natural killer; RAGE, receptor for advanced glycation end products; SQ-IHC, semi-quantitative immunohistochemistry; TNF-α, tumor necrosis factor alpha; TREM1, triggering receptor expressed on myeloid cells 1; TTR, transthyretin.

### 4.2. Mechanistic Insight from Animal Studies

Historically, most of the animal models for ATTRv amyloidosis were transgenic mice expressing human TTR variants [80]. The first transgenic mouse model was described by Yi and colleagues, using an inbred strain of mouse, C57BL/6, with an attempt to reproduce clinical and pathological features of amyloid polyneuropathy [81]. Currently, the most used transgenic mouse model is TTR/HSF1, lacking the main heat shock transcription factor (Hsf1). This leads to extensive fibrillar TTR deposition in several organs, including the peripheral nervous system [82]. The mechanism by which Hsf1 can protect from amyloid deposition is still uncertain because this transcription factor has multiple beneficial effects on proteostasis, either directly inhibiting TTR aggregation through specific chaperones, or protecting the target tissue from amyloid toxicity [80]. The studies investigating the role of the immune response in ATTR amyloidosis animal models are summarized in Table 2.

Nonetheless, when analyzing the studies exploring the role of the immune response in animal models, we have to admit that they are restricted to only a very limited number of research groups, and their results are certainly not directly exportable to humans.

Santos and colleagues studied the amyloid deposits in mice with the V30M variant in the sciatic nerve and dorsal root ganglia, which showed an increase in pro-inflammatory cytokines TNF-α and IL1-β, similar to human FAP nerves [82]. NF-kB activation occurred in dorsal root ganglia, and TNF-α and IL1-β increase exclusively related to TTR deposition in the peripheral nervous system, which suggests ongoing inflammation [82].

Gonçalves and colleagues showed a decrease in the frequency of innate immune cells (i.e., neutrophils and macrophages) in V30M-injured nerves of six-month-old V30M mice, as compared to wild-type mice nerves, with reduced expressions of chemokines, such as Cxcl-3, Cxcl-2, and Cxcl-12, and of Toll-like receptor (TLR) 1 in V30M mice [83]. Cxcl-2 is an essential chemokine for the recruitment of neutrophils, [84] Cxcl-3 is a chemokine important for the migration and adhesion of monocytes, [85] and Cxcl-12 contributes to neurite growth and axonal regeneration [86]. Moreover, they observed that after nerve injury, V30M mice had lower expressions and production of TNF-α and IL-1β compared to WT mice [83]. Finally, they found no difference in the expression of IL-6.

The absence of immune-inflammatory cellular infiltrate around TTR aggregates is thought to contribute to the lack of TTR clearance and, in turn, to worsen the disease [87,88]. Gonçalves and colleagues hypothesized that a non-optimal activation of Schwann cells after nerve injury, the decreased expression of TLR-1, and the reduction in pro-inflammatory cytokines and chemokines would deprive the damaged nerve of relevant factors for tissue regeneration. From this perspective, the peak of the anti-inflammatory cytokine IL-10 expression in V30M mice during the early phases after nerve injury may be responsible for the reduced immune activation in V30M mice.

Furthermore, Gonçalves’s group investigated the molecular mechanism behind this phenomenon. They performed a whole-mouse genome microarray analysis of injured V30M nerves, profiling more than 521 genes showing downregulation in several of the genes associated with immunity, including Tlr1, CXCL10, CXCL2, CCL2, CXCL3, CCL4, and CXCL11 [89].

Moreira and colleagues found that the expressions of Ccl20, Ccl8, Ccl5, Cxcl5, Ccl2, Cxcl2, and Cxcl3 were highly downregulated in the peripheral nervous system of a V30M TTR/HSF1 transgenic mouse model. They also documented that mouse Schwann cells stimulated with WT TTR secreted several chemokines, including CCL20, CXCL3, and CCL8, a process mediated by TLR-4 [90]. To further validate these results, the serum levels of Ccl20 and Cxcl2 were assessed via ELISA in wild-type TTR/HSF1 and V30M TTR/HSF1 mice at 6 and 20 months of age. A significant decrease in Ccl20 levels in the V30M TTR/HSF1 mice at 6 months of age compared to the wild-type TTR/HSF1 mice was observed, but no significant differences were detected in older V30M TTR/HSF1 mice. Moreover, contrary to what was affirmed by Gonçalves’s group, the expression of IL-6, induced in Schwann cells by wild-type TTR/HSF1 was significantly higher than that induced by V30M TTR/HSF1, while both TNF-α and IFN-β were poorly induced by either form of TTR [90].

A significant downregulation of the TLR-2 and TLR-4 signaling pathways has been demonstrated in bone marrow-derived macrophages from V30M TTR mice, with a consequent reduced expression of several chemokines as compared with wild-type ones [91].

Interestingly, Buxbaum and colleagues studied a transgenic model expressing approximately 90 copies of the wild-type human TTR gene under the control of its own promoter; this represented a good model, given that its tissue deposition patterns overlapped those observed in humans [92]. Transcriptomic analysis of both the liver and the target organs (heart and kidney) demonstrated that hepatic chaperone activity was deficient in mice with cardiac deposition, and that there was a robust cardiac inflammatory response in 3-month-old mice that had no cardiac deposits, which change in the hearts of 15–24-month-old mice with either fibrillar or non-fibrillar deposits. These genes mainly include those characterizing the innate immune response, but also those characteristics of T-cell activation [92].

The demonstration of inflammatory pathways in an FAP animal model was given by showing the effect of treatment with the IL-1 receptor antagonist Anakinra in FAP mice. It decreased inflammation markers and improved axonal non-myelinated fibers, demonstrating the role of inflammation in the pathogenesis of the disease. These results suggested the possible application of this drug in FAP patients [93].

**Table 2 cells-12-02383-t002:** Summary of selected studies on animal models of ATTRv amyloidosis evaluating the immune response modifications.

Reference	Animal Model	Methods and Techniques	Main Findings
Santos et al., 2010 [82]	V30M TTR/HSF1 mice vs. WT mice	SQ-IHC	Increase in pro-inflammatory cytokines TNF- α and IL1-β, and NF-kB activation occurring in dorsal roots ganglia.
Gonçalves et al., 2014 [83]	V30M TTR/HSF1 mice vs. WT mice	Flow cytometry and SQ-IHC	Downregulation of Cxcl-3, Cxcl-2, Cxcl-12, and TLR 1. Lower expressions of TNF-α and IL-1β. Upregulation of IL-10. No difference in the expression of IL-6.
Gonçalves et al., 2016 [89]	V30M TTR/HSF1 mice vs. WT mice	Microarray technology	TLR 1, Cxcl2, and Cxcl 3 were confirmed to be downregulated.
Moreira et al., 2021 [90]	V30M TTR/mice vs. WT mice	Real-time PCR	Decreased expressions of chemokines, such as Ccl20, Ccl8, Ccl5, Cxcl5, Ccl2, Cxcl2, and Cxcl3. Downregulation of IL-6.
Moreira et al., 2023 [91]	V30M TTR/mice vs. WT mice	Real-time PCR, cell culture	The expressions of several chemokines by bone marrow-derived macrophages generated from V30M TTR mice after stimulation with TLR4 and TLR2 agonists decreased; p38, which has a pivotal role for TLR4 and TLR2 signaling pathways, presented a reduced phosphorylation in V30M macrophages, compared to WT ones.
Gonçalves et al., 2015 [93]	V30M TTR/mice vs. WT mice	SQ-IHC; double immunofluorescence; immunogold labeling; real-time PCR; flow cytometry; Western blot; sciatic nerve morphometric analysis	Treatment with the IL-1 receptor antagonist Anakinra in FAP mice decreased inflammation markers and improved axonal non-myelinated fibers.
Buxbaum et al., 2012 [92]	Transgenic model expressing approximately 90 copies of the wild-type human TTR gene under the control of its own promoter	Transcriptomic analysis	Hepatic chaperone activity was deficient in mice with cardiac deposition; robust cardiac inflammatory response in 3-month-old mice who have no cardiac deposits, which changes in the hearts of 15–24-month-old mice with either fibrillar or non-fibrillar deposits.

ATTR, transthyretin amyloidosis; Ccl, chemokine ligand; Cxcl, C-X-C motif chemokine ligand 1; HSF1, heat shock factor 1; IL, interleukin; NF-kB, nuclear factor kappaB; PCR, polymerase chain reaction; SQ-IHC, semi-quantitative immunohistochemistry; TLR, Toll-like receptor; TNF-α, tumor necrosis factor-alpha; TTR, transthyretin; WT, wild type.

## 5. Future Perspectives

The characterization of the role of the immune response in FAP is still in the early stages; adequate, multicenter studies carried out on a large number of patients are needed. From this perspective, an adequate assessment of local and systemic inflammatory changes in a large number of patients harboring different mutations may help in adequately defining contributions of the immune response to disease progression. In our opinion, it is also important to define whether the immune response is simply induced by the deposition of amyloid fibrils, or if it directly intervenes in the increase in the deposition of the fibrils. Moreover, it is necessary to distinguish between an antibody-mediated response, which may have a protective role in removing the fibrils, and the inflammatory response in general, which still needs to be characterized in terms of consequences. Finally, we should acknowledge that there is a lack of adequate longitudinal studies on ATTRv patients that can adequately describe and understand how the immune response develops and varies in the different phases of the disease. In fact, we believe that the evolution of the immune response during the course of the disease could be the key to fully understanding its role.

## 6. Conclusions

Accumulating evidence suggests that the immune response may play some role in ATTRv amyloidosis, with multiple possible effects on disease onset and progression. These effects are understood only in part, which current and future studies will clarify and hopefully form the basis for future drug trials.

## Figures and Tables

**Figure 1 cells-12-02383-f001:**
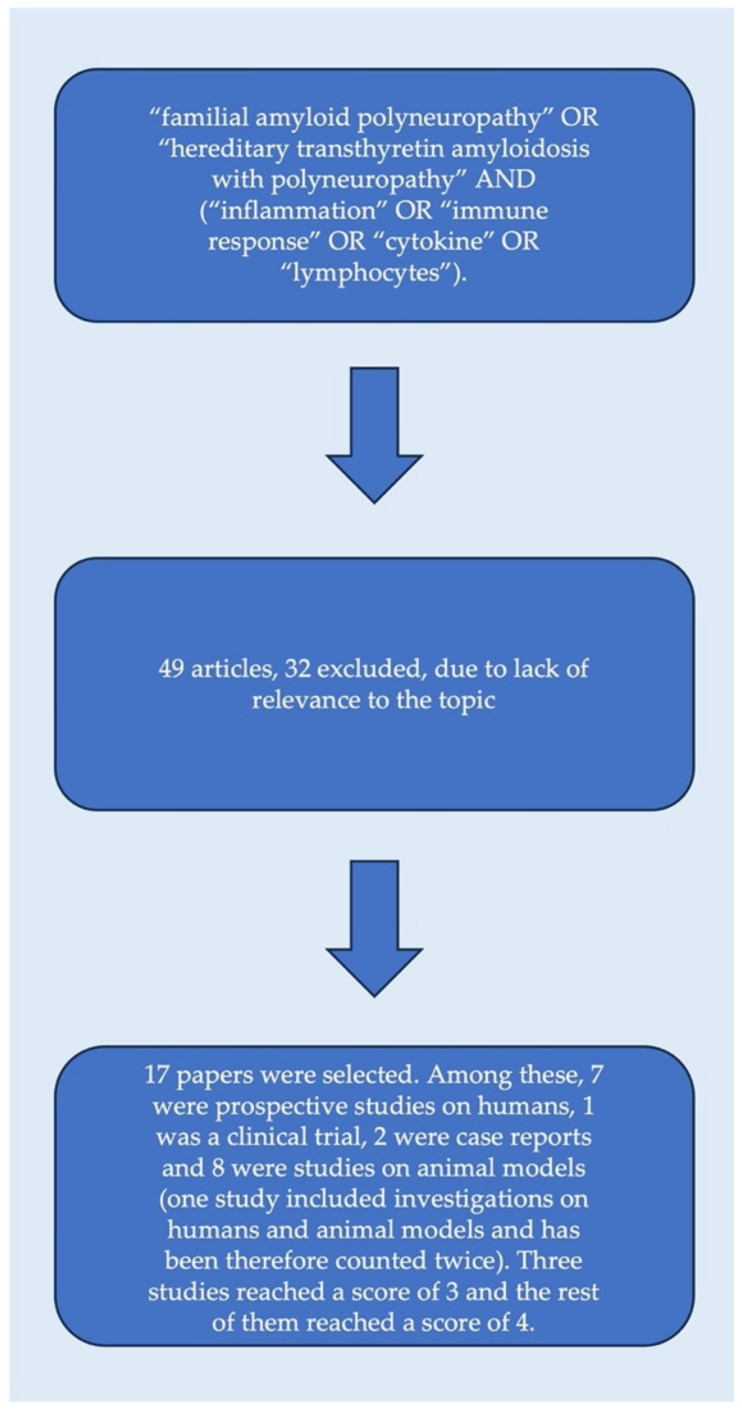
Box diagram that summarizes the selection process of the studies.

**Figure 2 cells-12-02383-f002:**
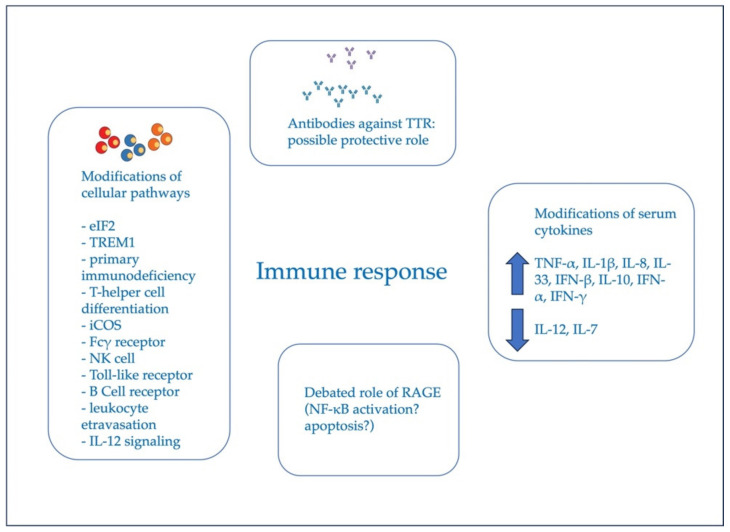
This figure summarizes the possible role of the immune response in ATTRv amyloidosis.

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
