# Peer review of "Current Evidence Supporting the Role of Immune Response in ATTRv Amyloidosis"

_cells, 2023, doi:10.3390/cells12192383_

Round 1
Reviewer 1 Report (Previous Reviewer 2)
It is the obligation of a review to both cite the relevant literature and then critically assess it, particularly in instances in which a variety of studies of the same phenomenon are not consistent or in conflict with each other. Further it is often helpful for the reader if the author can propose a model which encompasses most of the observations and can be tested by further experiments.
In the abstract the authors state:
“In fact, the immune response may be involved in the development and progression of the disease. Fibrillary TTR species bind to the receptor for advanced glycation end products (RAGE), probably activating the nuclear factor κB (NF-κB) pathway. Moreover, peripheral blood levels of several cytokines, including, interferon (IFN)-gamma, IFN-alpha, IL-6, 21 IL-7, and IL-33 are altered in the course of the disease. “
These presumably represent the facts upon which their hypothesis rests when they state:
“It has been postulated that the immune response may participate in the disease's development and progression; recent studies are exploring this hypothesis with interesting results.”
They seem to ignore the possibility that, as in infectious diseases, the immune/inflammatory response may be an attempt at host protection and do not give much credence to the notion proposed in their reference 93, and the recent report of the resolution of cardiac TTR deposition in a small number of patients whose plasma contained a TTR specific antibody (Fontana et al N Engl J Med. 2023 Jun 8;388(23):2199-2201. doi: 10.1056/NEJMc2304584.PMID: 37285532 ). In the first the authors propose a successful response to either the misfolded or fibrillar TTR which differs in the transgenic mice which do not show deposition or in the second a small number of patients in whom documented bona fide tissue deposition is resolved over time, presumably as an effect of a successful antibody response. While it is true in both these instances the predominantly affected organ was the heart, most of the data in the field consider the process similar or identical regardless of the main clinical manifestations. If the authors do not consider this to be the case, then the counter argument should be included in their discussion.
With respect to the text:
Lines 67-140: Briefly summarize the background on TTR polyneuropathy. We have previously commented on those statements that do not accurately reflect our current state of knowledge. Interestingly 2 of the 3 examples of ATTRv cited, i.e. V30M, ala60 have mixed polyneuropathic/cardiomyopathic phenotypes.
While ref 17 is cited as the source for dorsal root and sensory ganglia being the major site of deposition, other studies e.g. Koike, indicate dominant deposition along the path of the nerve impacting on Schwann cells and neurons.
Given the worldwide frequency of V122I as well as the mixed phenotype of V30M cardiomyopathies are probably the most common phenotypes of ATTR v. While ATTR V30M was originally described as a peripheral neuropathy, it is very clear now that cardiomyopathy is a common feature, particularly in cases of late onset, i.e. non-endemic.
Given this statement confining the literature review to papers discussing “polyneuropathy” is limiting and incorrect. While the toxic processes in the peripheral nerve and heart may not be precisely the same, although tissue culture studies of cytotoxicity suggest that they are similar, it is likely that the process of TTR synthesis by the liver and its time in the circulation are very similar. Hence it is also likely that the systemic (? inflammatory) response is the same.
Table 1: You do not discuss the discrepancy in the RAGE analyses regarding activation of NFκB in the Sousa and Matsunaga studies. In lines 162-176 you discuss the NFκB pathways in detail as if those two studies agree on the data with respect to RAGE and NFκB. Why are these reports discrepant? Is it merely a difference among tissues or is there something wrong with one of the data sets? Do you believe the NFκB results since the two studies were not confirmatory? Say so and why you have come to your conclusion. Are there other studies that support either view?
Lines 177-192: you discuss the non-specificity of RAGE binding across many neurologic disorders and different forms of amyloid and it is not clear from the literature that, even if this is true, it is not TTR polyneuropathy specific. In fact, published data suggest a neuronal target in many forms of neuronal pathology, particularly diabetic and some inflammatory states, not just FAP. The interaction of RAGE with fibrils may be coincident with the toxic response, independent of tissue damage. Its role requires much more discussion. For instance, did the studies talking about RAGE in FAP demonstrate or discuss a role for s100B, or any of the other S100 proteins. You do not seem to come to any conclusion regarding the relevance or its specificity in the case of the response to TTR aggregates. Is this FAP specific or not????
In examining the various papers cited in table 1, what cytokines are common among the various reports? As best I can tell none. The only common features appear to be TNFα and IL1β in 2 studies (Sousa et al and Azevedo et al), both of which have the Saraiva group as participants. By the way references 7 and 75 are the same.
Similarly in table 2 it would be useful in the discussion to note which inflammatory indicators are shared and which are not and stress the studies in which serial assays show process dependent changes suggesting a sequence of events that could be summarized in the model; e.g. does non-immunologically based inflammation result in formation of new epitopes that are recognized by either the humoral or cell-based immune systems, which in turn make the tissue damage better (as suggested by the Fontana NEJM paper) or worse, or does the presence of a misfolded TTR stimulate a salutary or tissue damaging response?
It is the hope of the reader such a review that the authors will put together a potential model against which further observations can be judged to either support the model or lead to the creation of a new and more informative/insightful picture of the process of the host response to a misfolded amyloidogenic protein, in this case TTR.

Author Response
see pdf
Reviewer 2 Report (New Reviewer)
The authors present a review of the literature about the role of the immune response in ATTRv amyloidosis. Most relevant articles have been found and presented in a clear way.
Comments:
- The evidence for inflammation in humans with ATTRv amyloidosis is not substantial. There are good arguments for a role of FAS, AGE and RAGE. They seem to be relevant in ATTRv amyloidosis but their effect is more related to toxicity with secondary some inflammatory changes (similar to phenomena as necrosis) than inducing manifest clinical inflammation. The authors describe upregulation of some inflammatory proteins, but clear clinical signs of inflammation or the immune response such as elevated CRP and the presence of inflammatory cells in biopsies is lacking in humans. It should be made clear in the text that really convincing evidence for inflammation or the immune response in human beings suffering from ATTRv amyloidosis is still lacking.
- The animal studies presented in table 2 are all from the group of Maria Saraiva except the one of Joel Buxbaum about chaperone proteins. Although I am fascinated by their work for many years, animal studies are not human studies and inflammation (with the favorable effect of Anakinra) in animals does not convince that a similar phenomenon is also present in humans. Mice are not (wo)men! So, the authors should state in the text the limitation that table 2 is restricted to only one/two research groups.
- In Future perspectives line 368-370 speculations are made about drug trials in which new molecules can be identified for the treatment of ATTRv amyloidosis. It is hard to imagine based on the presented literature what kind of drug can be expected to fulfill more than an ancillary role. Even in the most favorable situation a drug like Anakinra might only mitigate some of the disease manifestations, but the disease process of amyloid accumulation is not expected to be stopped. The authors should either remove this statement or provide really convincing arguments for it.
- The review does not provide any convincing evidence for a central role of inflammation or the immune response in the development or progression of ATTRv amyloidosis in human beings. Therefore, in the Abstract (line 17) …, the central role of inflammation … must be formulated realistically and become …, a role of inflammation … .
- And in line 18 … In fact, the immune response may be … must be changed into … The immune response might be …
- And in the Introduction (line 39) … regarding the pivotal role of … should be changed into … regarding some role of …
- Similarly in the conclusions (line 372) … plays a significant role … is not backed by the results that have been discussed and should be modified into … may play some role …
- It would be helpful for the interested reader if in the tables the reference numbers are also presented. Where can I find Mendez Sousa 2001?
Author Response
see pdf

Reviewer 3 Report (New Reviewer)
This is a well written paper and should be published
Minor recommendation
1. On Materials and methods: this needs to be expanded. Recommend a Box Diagram : including the search items with how many articles first found; then subsequent exclusions ( what were exluded eg case reports, short articles etc) with how many articles left; then how many articles were reviewed for relevance ( listing the relevance e.g 0: not relevant, 1: minor……. 5 very relevant): then list final # of articles included in this review.
For the final articles: important to note how many were: prospective studies, restrospective studies, reviewes etc.
2. Line 110--- please add Vutrisiran TTR gene-selencing med, as this is FDA approved (David Adams, Amyloid 2023 Mar;30(1):1-9)
Author Response
see pdf

Reviewer 4 Report (New Reviewer)
The manuscript titled "Current Evidence Supporting the Role of Immune Response in ATTRv Amyloidosis" submitted by Domenico Plantone and coworkers dissects the role of immune responses on Hereditary transthyretin (ATTRv) amyloidosis. This disease is a progressive and multisystemic disease, caused by up to 140 certain pathogenic variants described so far in the TTR gene, which has long been perceived as non-inflammatory due to the absence of mononuclear cell infiltration in ex vivo tissues. However, recent research has unveiled the pivotal role of inflammation in its pathogenesis. Indeed, the fibrillary TTR species, central to the disease, are shown to interact with the receptor for advanced glycation end products (RAGE), potentially activating the nuclear factor κB (NF-κB) pathway. Importantly enough, the review also highlights the importance of alterations in peripheral blood levels of key cytokines, including interferon (IFN)-gamma, IFN-alpha, IL-6, IL-7, and IL-33 throughout the course of the disease. In this context, this review also explores the current state of the art of potential therapeutic approaches as well future perspectives in the field.
Overall, the manuscript is well-written and provides valuable insights into the immune response's role in ATTRv amyloidosis, bridging the gap between past and current findings. Moreover, the authors have employed a comprehensive and well-structured approach in gathering relevant research, contributing to the robustness of their analysis.
In my opinion, the manuscript is suitable for publication in cells in present form.
Only minor corrections of the English language is needed.
Author Response
see pdf

Round 2
Reviewer 1 Report (Previous Reviewer 2)
Review: Current evidence supporting the role of immune response in ATTRv amyloidosis
It is the obligation of a review to both cite the relevant literature and then critically assess it, particularly in instances in which a variety of studies of the same phenomenon are not consistent or in conflict with each other. Further it is often helpful for the reader if the author can propose a model which encompasses most of the observations and can be tested by further experiments.
In the abstract the authors state:
“In fact, the immune response may be involved in the development and progression of the disease. Fibrillary TTR species bind to the receptor for advanced glycation end products (RAGE), probably activating the nuclear factor κB (NF-κB) pathway. Moreover, peripheral blood levels of several cytokines, including, interferon (IFN)-gamma, IFN-alpha, IL-6, 21 IL-7, and IL-33 are altered in the course of the disease. “
These presumably represent the facts upon which their hypothesis rests when they state:
“It has been postulated that the immune response may participate in the disease's development and progression; recent studies are exploring this hypothesis with interesting results.”
They seem to ignore the possibility that, as in infectious diseases, the immune/inflammatory response may be an attempt at host protection and do not give much credence to the notion proposed in their reference 93, and the recent report of the resolution of cardiac TTR deposition in a small number of patients whose plasma contained a TTR specific antibody (Fontana et al N Engl J Med. 2023 Jun 8;388(23):2199-2201. doi: 10.1056/NEJMc2304584.PMID: 37285532 ) in which the authors propose a successful response to either the misfolded or fibrillar TTR which differs in the transgenic mice which do not show deposition or a small number of patients in whom documented bona fide tissue deposition is resolved over time, presumably as an effect of a successful antibody response. While it is true in both these instances the predominantly affected organ was the heart, most of the data in the field consider the process similar or identical regardless of the main clinical manifestations. If the authors do not consider this to be the case, then the counter argument should be included in their discussion.
With respect to their text:
Lines 67-140: Briefly summarize the background on TTR polyneuropathy. We have previously commented on those statements that do not accurately reflect our current state of knowledge. Interestingly 2 of the 3 examples of ATTRv cited, i.e. V30M, ala60 have mixed polyneuropathic/cardiomyopathic phenotypes.
While ref 17 is cited as the source for dorsal root and sensory ganglia being the major site of deposition, other studies e.g. Koike, indicate dominant deposition along the path of the nerve impacting on Schwann cells and neurons.
Given the worldwide frequency of V122I as well as the mixed phenotype of V30M cardiomyopathies are probably the most common phenotypes of ATTR v. While ATTR V30M was originally described as a peripheral neuropathy, it is very clear now that cardiomyopathy is a common feature, particularly in cases of late onset, i.e. non-endemic.
Given this statement confining the literature review to papers discussing “polyneuropathy” is limiting and incorrect. While the toxic processes in the peripheral nerve and heart may not be precisely the same, although tissue culture studies of cytotoxicity suggest that they are similar, it is likely that the process of TTR synthesis by the liver and its time in the circulation are very similar. Hence it is also likely that the systemic (? inflammatory) response is the same.
Table 1: You do not discuss the discrepancy in the RAGE analyses regarding activation of NFκB in the Sousa and Matsunaga studies. In lines 162-176 you discuss the NFκB pathways in detail as if those two studies agree on the data with respect to RAGE and NFκB. Why are these reports discrepant? Is it merely a difference among tissues or is there something wrong with one of the data sets? Do you believe the NFκB results since the two studies were not confirmatory? Say so and why you have come to your conclusion. Are there other studies that support either view?
Lines 177-192: you discuss the non-specificity of RAGE binding across many neurologic disorders and different forms of amyloid and it is not clear from the literature that, even if this is true it is not TTR polyneuropathy specific. In fact, published data suggest a neuronal target in many forms of neuronal pathology, particularly diabetic and some inflammatory states, not just FAP. The interaction of RAGE with fibrils may be coincident with the toxic response, independent of tissue damage. Its role requires much more discussion. For instance, did the studies talking about RAGE in FAP demonstrate or discuss a role for s100B, or any of the other S100 proteins. You do not seem to come to any conclusion regarding the relevance or its specificity in the case of the response to TTR aggregates. Is this FAP specific or not????
Lines 173-191:The discussion does not address these issues, hence the subsequent discussion regarding NF-κB may not be relevant as suggested by their reference 53.
In examining the various papers cited in table 1, what cytokines are common among the various reports? As best I can tell none. The only common features appear to be TNFα and IL1β in 2 studies (Sousa et al and Azevedo et al), both of which have the Saraiva group as participants. By the way references 7 and 75 are the same.
Similarly in table 2 it would be useful in the discussion to note which inflammatory indicators are shared and which are not and stress the studies in which serial assays show process dependent changes suggesting a sequence of events that could be summarized in the model e.g. does non-immunologically based inflammation result in formation of new epitopes that are recognized by either the humoral or cell-based immune systems, which in turn make the tissue damage better (as suggested by the Fontana NEJM paper) or worse, or does the presence of a misfolded TTR stimulate a salutary or tissue damaging response?
It is the hope of the reader such a review that the authors will put together a potential model against which further observations can be judged to either support the model or lead to the creation of a new and more informative/insightful picture of the process of the host response to a misfolded amyloidogenic protein, in this case TTR.
New line 117-140: None of the hypotheses proposed necessarily eliminates the immune inflammatory response as playing a role in pathogenesis, in fact, any or all of them could induce provide the stimulus for such a response, which, instead of being salutary, increases neuronal death perhaps by increasing oxidative stress and subsequent apoptosis.
Line 247 “hypothesize” is misspelled.
Line 259: Fontana now reports on anti-TTR antibodies in ATTRv patients new reference 77, lines 286-296 which are inconsistent with statement in line 259. Further both O’nuallain and Michalon have reported isolating TTR specific B-cells from humans without ATTR amyloidosis, observations consistent with humans making such antibodies which may be responsible for those individuals not having ATTR at all or having subclinical amyloid. Again suggesting that the immune/inflammatory responses being observed are secondary to some initial event rather than being primary.
The reader would be well served by a figure depicting a potential model(s) for the involvement of immune/inflammatory responses in FAP and all systemic TTR amyloid pathogenesis. The present discussion is confusing to the reader since the data are not consistent among all the studies and are merely summarized without critical comment as to why one interpretation is better than another.
Needs some editing for nuances.
Author Response
Response to reviewer #1:
Point 1: Lines 173-191:The discussion does not address these issues, hence the subsequent discussion regarding NF-κB may not be relevant as suggested by their reference 53.
Response 1: As admitted by reviewer #1 her/himself, the debate regarding the role of RAGE and the possible activation of NF-kB is open and the methodological diversity of the studies does not, to date, allow us to reach any definitive conclusion without the risk of being contradicted. Therefore we believe that it is more cautious and respectful for the reader to describe the evidence regarding this subject, without necessarily taking sides. Considering the opinions of the different reviewers that commented this paper it is clear that there is no consensus even on the actual role of immune response in ATTRv and we need to respect this (see the previous critiques from reviewer #2). There is an ongoing debate on the non-specificity of RAGE binding across many neurologic disorders and different forms of amyloid and it is still not clear from the literature if this is TTR polyneuropathy-specific.
However, we agree on the need to highlight that RAGE activation appears to be nonspecific. Therefore, following the reviewer's suggestions, we have modified the text of the (new) lines 203-230 as follows:
“There is an ongoing debate on the non-specificity of RAGE binding across many neurologic disorders and different forms of amyloid and it is still not clear from the literature if this is TTR polyneuropathy-specific. In fact, published data suggest that the interaction of RAGE with fibrils may be coincident with the toxic response, independent of tissue damage. RAGE represents a member of the multiligand cell surface immunoglobulin family [55] and is expressed by different cells including endothelial and smooth muscle cells, dendritic cells, lymphocytes, neutrophils, monocytes, and macrophages.[56] The pivotal role of RAGE has already been documented in the pathogenesis of multiple neurological diseases including Alzheimer’s disease, amyotrophic lateral sclerosis, Parkinson’s disease, and Huntington’s disease. [57] Remarkably, RAGE has other ligands than AGE, including amphoterin,[58] S100/calgranulins,[59] and amyloids forming β-sheet fibrils.[60,61] Relevantly, the role of RAGE has also been investigated in the pathogenesis and progression of peripheral neuropathy. By binding its ligand s100B, which is secreted by the Schwann cells, RAGE mediates peripheral nerve repair in vivo.[62] However, RAGE, together with its pro-inflammatory ligands, has been found overexpressed in human diabetic nerves and the activation of the RAGE-NF-κB-dependent pathway seems to significantly contribute to the progression of the disease.[63] In further support of this evidence, it should also be remembered that both AGE and RAGE levels measured in skin biopsies strongly correlate with neuropathy severity.[64,65] Finally, Yan and colleagues in a mouse model of systemic amyloid A (AA) amyloidosis, demonstrated that RAGE also binds amyloid A, and that by antagonizing RAGE, NF-κB activation is suppressed, as well as the expression of proinflammatory cytokines, and accumulation of amyloid.[66] From this perspective, a recent study found decreased S100A8 plasma levels in ATTRV30M patients, as well as a dysregulated S100 expression by Schwann cells in response to TTRV30M and by mutated macrophages in response to toll like receptors agonists.[67] Taking into account the data from the literature, we would be inclined to define the role of RAGE as non-ATTRv-specific.”
We hope that the reviewer agrees that this allows us to keep together all the opinions regarding the topic, better clarifying the message that she/he advised us to highlight.
Point 2: New line 117-140: None of the hypotheses proposed necessarily eliminates the immune inflammatory response as playing a role in pathogenesis, in fact, any or all of them could induce provide the stimulus for such a response, which, instead of being salutary, increases neuronal death perhaps by increasing oxidative stress and subsequent apoptosis.
Response 2: We agree with the reviewer and therefore we added the following paragraph (lines 152-155):
“Nevertheless, taking into account all the possible pathogenetic hypotheses previously described, it must be admitted that none of them necessarily eliminates the immune inflammatory response as playing a role in the development of the disease. In fact, any or all of them could induce or provide the stimulus for immune activation.”
Point 3: Line 247 “hypothesize” is misspelled.
Response 3: The spelling has been corrected.
Point 4: Line 259: Fontana now reports on anti-TTR antibodies in ATTRv patients new reference 77, lines 286-296 which are inconsistent with statement in line 259. Further both O’nuallain and Michalon have reported isolating TTR specific B-cells from humans without ATTR amyloidosis, observations consistent with humans making such antibodies which may be responsible for those individuals not having ATTR at all or having subclinical amyloid. Again suggesting that the immune/inflammatory responses being observed are secondary to some initial event rather than being primary.
Response 4: We thank the reviewer for this critique.
We modified line 259 as follows: “Furthermore, no data is available on the circulating peripheral blood mononuclear cell modifications in ATTRv patients.”
We also added the following paragraph (new lines 315-326) as suggested by reviewer#1:
Moreover, other Authors reported TTR specific B-cells from humans without ATTR amyloidosis.[78,79] These observations reinforce the hypothesis that these antibodies could be involved in the removal of amyloid in the presymptomatic phase of the disease in some subjects not having ATTR at all or having subclinical amyloid. This evidence suggests that this B-cell response is likely secondary rather than primary in the disease pathogenesis. Interestingly, Michalon and colleagues identified NI301A, a monoclonal antibody that selectively binds with high affinity to the disease-associated ATTR aggregates of either wild-type or ATTRv-related disease. This process occurred through the isolation of a collection of ATTR-binding antibodies selected for high-affinity binding to ATTR, amyloid-removal activity, and absent binding to naturally folded TTR, by screening human memory B-cell libraries derived from healthy elderly subjects.[78]
Point 5: The reader would be well served by a figure depicting a potential model(s) for the involvement of immune/inflammatory responses in FAP and all systemic TTR amyloid pathogenesis. The present discussion is confusing to the reader since the data are not consistent among all the studies and are merely summarized without critical comment as to why one interpretation is better than another.
Response 5: Reviewer #1 writes to us: "The present discussion is confusing to the reader since the data are not consistent among all the studies and are merely summarized without critical comment as to why one interpretation is better than another." We agree with the reviewer that the literature data are not uniform. And even considering the opinions of the different reviewers that commented this paper it is clear that there is no consensus (see the previous critiques from reviewer #2). We reiterate that we do not want to take a position at all costs on the protective or detrimental role of inflammation in ATTRv or whether the involvement of one inflammatory pathway reported by one research group is more relevant compared to another. And we do this not out of pure inability to take a position, but out of respect for the evidence present in the literature so far. We believe it is now clear that it is necessary to distinguish between antibody-mediated response and inflammatory response in the general sense. Furthermore, we have clearly explained in the "future directions" paragraph that it will be necessary in the future "...to define whether the immune response is simply induced by the deposition of amyloid or if it directly intervenes in the increase of the deposition of the fibrils." In fact, there is a lack of adequate longitudinal studies on patients that can adequately describe and understand how the immune response develops and varies in the different phases of the disease.
However, taking into account the criticism of reviewer #1 we have added the following part:
Moreover, it is necessary to distinguish between antibody-mediated response, which may have a protective role in removing the fibrils, and the inflammatory response in general which still needs to be characterized in terms of consequences. Finally, we should acknowledge that there is a lack of adequate longitudinal studies on ATTRv patients that can adequately describe and understand how the immune response develops and varies in the different phases of the disease. In fact, we believe that the evolution of the immune response during the course of the disease could be the key to fully understanding its role.
Finally, figure 2 has been added to summarise the role of the immune response in ATTRv.
English has been reviewed by a native speaker.
All the new changes are highlighted in blue.

This manuscript is a resubmission of an earlier submission. The following is a list of the peer review reports and author responses from that submission.
Round 1
Reviewer 1 Report
As per file

English good
Reviewer 2 Report
Lines 74-75: “In relation to the TTR mutations, over 140 causative mutations have been identified so far, with Val30Met being the most common, especially in Japan, Portugal, and Cyprus”.This is incorrect. Given its distribution in the U.S. and sub-Saharan Africa it is likely that TTR V122I is the most common disease associated mutation worldwide. V30M may be the most common mutant allele associated with polyneuropathy, but it is not the most common amyloid related variant (1.5 million carriers in U.S. alone, with age-dependent penetrance.
Lines 79-80: “ATTRv amyloidosis is generally regarded as a multisystem disorder, with the peripheral nervous system and heart being the two main sites of involvement”.
Given this statement confining the literature review to papers discussing “polyneuropathy” is limiting and incorrect. While the toxic processes in the peripheral nerve and liver may not be precisely the same, although tissue culture studies of cytotoxicity suggest that they are similar, it is likely that the process of synthesis by the liver and its time in the circulation are very similar and it is likely that the systemic (?inflammatory) response is the same. While ATTR V30M was originally described as a peripheral neuropathy, it is very clear now that cardiomyopathy is a common feature, particularly in cases of late onset, i.e. non-endemic.
Lines 114-120: It appears from most tissue culture studies that TTR oligomers, not the fibrils are not the toxic elements in any target organ. Further there are in vivo studies in humans (Saraiva) that report the appearance of non-fibrillar deposits appearing in both peripheral nerves of early V30M patients, as well as in human wtTTR transgenic mice showing non-fibrillar cardiac deposits prior to fibril deposition, although this has not yet, to my knowledge been reported in human TTR cardiomyopathy. It is clear from tissue culture studies that TTR oligomers induce oxidative stress as measured by MTT reduction or caspase activation (Reixach and Manral and Reixach).
Table 1: You do not discuss the discrepancy in the RAGE analyses regarding activation of NFκB in the Sousa and Matsunaga studies. In lines 162-176 you discuss the NFκB pathways in detail as if those two studies agree on the data with respect to RAGE and NFκB. Why are these reports discrepant? Is there something wrong with one of the data sets?
Lines 177-192: you discuss the non-specificity of RAGE binding across many neurologic disorders and different forms of amyloid and it is not clear that this is TTR polyneuropathy specific. In fact, the data suggest a neuronal target in many forms of neuronal pathology, not just amyloids. The interaction of RAGE with fibrils may be coincident with the toxic response independent of tissue damage. Its role requires much more discussion. For instance, did the studies talking about RAGE in FAP demonstrate or discuss a role for s100B, or any of the other S100 proteins.
Lines 199-221: You do not mention the work of Kurian et al who studied peripheral blood transcriptomics in 263 Portuguese V30M subjects and Luminex bead assays in 40 of those subjects, a body of data far greater than the combined data in the three studies you cite.
Lines 246-297: Your analysis assumes that the HSF1 heterozygous knockouts are a good model to look at the process of amyloidogenesis and the host response to it in vivo. While it does allow the formation of tissue deposits, they do not, as I recall, have a peripheral neuropathy phenotype. A much more informative study, regarding the pathogenesis of human TTR deposition is a 2016 study in which transgenics were generated by over-expression of the wild type human TTR gene with bona fide cardiac amyloid deposition and transcriptomic analysis of both the liver and the target organs (heart and kidney) demonstrating that hepatic chaperone activity was deficient in mice with cardiac deposition and that there was a robust cardiac inflammatory response in 3 month old mice, who have no cardiac deposits which changes in the hearts of 15-24 month old mice with either fibrillar or non-fibrillar deposits, a much more robust set of observations supporting an important role for inflammation in the pathogenesis of tissue deposition. Granted that this was a cardiomyopathy model it is likely that it shares much in the way of host response to a neuropathic challenge. If your literature search had been broader, i.e. looking for systemic TTR amyloidosis, you might have found these observations more relevant to your hypothesis.

Round 2
Reviewer 1 Report
Abstract and Introduction need revision for the English.
References need substantial revision in the text and the bibliography list. There is a tremendous mistake here. It is non-sense
Cardoso et al., was not taken out from table 2 as the authors refer in their answer to reviewer 1.
English adequate except for abstract and introduction
Author Response
The whole manuscript, including the abstract and introduction, has been revised by a native English speaker.
The whole doxycycline part, including the Cardoso et al. reference, was taken out from the text and from the tables.